# Frequency Selective Surface for Ultra-Wide Band Filtering and Shielding

**DOI:** 10.3390/s22051896

**Published:** 2022-02-28

**Authors:** Aldo De Sabata, Ladislau Matekovits, Adrian Buta, Gianluca Dassano, Andrei Silaghi

**Affiliations:** 1Department of Measurements and Optical Electronics, Politehnica University Timisoara, 300006 Timisoara, Romania; aldo.de-sabata@upt.ro (A.D.S.); ladislau.matekovits@polito.it (L.M.); adrian.buta@student.upt.ro (A.B.); 2Department of Electronics and Telecommunications, Politecnico di Torino, 10129 Turin, Italy; gianluca.dassano@polito.it; 3Istituto di Elettronica e di Ingegneria dell’Informazione e delle Telecomunicazioni, National Research Council, 10129 Turin, Italy

**Keywords:** frequency selective surface, FR4, spatial band-stop filter, ultra-wide band (UWB), circuit model

## Abstract

A frequency selective surface for spatial filtering in the standardized Ultra-Wide Band (UWB) frequency range is proposed. A very large stop-band of 1.75–15.44 GHz has been obtained, with good polarization insensitivity and an angular stability of more than 60${}^{\circ }$ and more than 50${}^{\circ }$ in TE and TM incidence, respectively. Circuit models have been devised. The structure has been assessed by electromagnetic simulation and implemented on an FR4 substrate of 1.6 mm thickness, with an edge of the square-shaped unit cell of 15 mm. Tests in an anechoic chamber demonstrated good matching between simulation and experimental results and proper operation of the device.

## 1. Introduction

Frequency selective surfaces (FSSs) implemented on single-layer or stacked printed circuit boards (PCBs) [1,2] have found multiple applications in the last decade as radomes, absorbers, polarizers, artificial magnetic conductors, spatial filters and shields, dichroic reflectors and reflectors for antenna gain enhancement, etc.

With the advent and exponential growth of wireless communications technology in the last two decades, a special interest in applications of FSSs as spatial filters, screens and shields has emerged. Screens can be deployed to protect certain spaces, including rooms and buildings, from electromagnetic waves in some frequency bands, while leaving unaffected signals from different frequency bands. Electronic high frequency circuits and equipment can be shielded in selected frequency bands by embedding them into specially designed boxes with patterned walls. Such functionalities are analog to the radomes one, which allows the antenna to communicate at the operational frequency while stopping unwanted signals. Angle independence and polarization insensitivity of the frequency response to incident electromagnetic waves are required for these kinds of applications [3]. Other more specific applications have also been reported, as for example in imaging [4] or in remote sensing in [5].

Several works have been devoted to spatial filtering in some standardized wireless communication frequency bands by means of FSSs. A single layer, double-metallization solution, built on FR4 substrate that filters electromagnetic signals from WiMAX, WLAN, and X-band has been proposed in [6]. The miniaturized unit cell (10 × 10 mm${}^{2}$) was based on combinations of square loops and dipoles. The structure has been tested up to a 50${}^{\circ }$ angle of incidence.

A geometry based on a modified circular loop that introduced two notch frequencies in the ISM band at 2.4 and 5.8 GHz has been introduced in [7] in view of mitigating the mutual interference of indoor adjacent wireless networks. An optimization technique has been proposed to allow for the independent adjustment of the two frequencies. The structure offered stable frequency response up to a 30${}^{\circ }$ angle of incidence. Filtering in the GSM bands has been considered in [8]. A proposed square unit cell with a side of 50 mm and consisting of a combination of a square loop resonator and a more complicated loop-shaped resonator has been demonstrated to filter signals at 942, 1842, and 2142 MHz, up to a 45${}^{\circ }$ angle of incidence. The small dimension experimental prototype was realized on an FR4 substrate and contained 4 × 4 elements.

GSM shielding has been approached again in [9], where a 2.5D structure with two frequency notches at 900 and 1790 MHz has been proposed. The unit cell was based on a modified double-square loop element knitted on both sides of a PTFE substrate, and interconnecting vias, for a dimension of the square-shaped unit cell of 24 mm. Correct operation has been demonstrated up to an angle of incidence of 60${}^{\circ }$. A number of five notches are introduced by the FSS solution proposed in [10], covering the GSM, GPS, Bluetooth/Wi-Fi, Wi-MAX and WLAN bands, designed in view of shielding against smartphone and tablet electromagnetic signals. The unit cell consisted of five square and modified square rings imprinted on an FR4 substrate, for dimensions of the unit cell of 27 × 25.7 mm${}^{2}$. The structure has been experimentally tested for incidence angle stability up to an angle of 45${}^{\circ }$.

Shielding by incorporation of FSS into building elements has been considered in [11]. A 20 × 20 mm${}^{2}$ unit cell has been designed, based on circular and elliptical loops and deposited on window glass, affecting transparency only by an amount of 14%. The prototype, containing 13 × 10 unit cells and operating as a band-stop filter at 2.45 and 5.5 GHz (Wi-Fi and WLAN), has been demonstrated to be polarization insensitive and angle independent up to 45${}^{\circ }$. The interest in stop-band filters for wireless applications is demonstrated also by the conception of an active surface containing p-i-n diodes as control elements [12], with a unit cell based on ring patches and cross-dipoles, and exhibiting closely spaced resonant frequencies of 2.36 and 2.89 GHz. The proposed structure, with dimensions of the unit cell of 21 × 18 mm${}^{2}$, has been implemented on an FR4 prototype board with dimensions of 20 × 20 cm${}^{2}$ and tested in normal and oblique incidence up to an angle of 40${}^{\circ }$.

The evolution of the wireless technology raised the interest in research for wide-band solutions too. An early design, with a large –10 dB stop-band in the interval 6.5–14 GHz, has been reported in [13]. The unit cell, with dimensions 12 × 12 mm${}^{2}$, was based on a combination of a cross dipole on one face of the PCB and a circular ring on the other one. The prototype, comprising 36 × 23 unit cells, has been implemented on an FR4 substrate of 3.2 mm thickness, demonstrating functionality up to an angle of incidence of 45${}^{\circ }$.

A solution for shielding in the X band, with a central frequency around 10 GHz and a fractional bandwidth (Δf/$f_{0}$) of 48% has been reported in [14]. The unit cell pattern was based on convoluted square loops for better polarization stability. The edge of the square-shaped unit cell was of 6.8 mm. Two prototypes have been fabricated, one on a single-sided Rogers/Duroid flexible laminate with a thickness of 0.127 mm in view of conformal applications, and one on an FR4 substrate with thickness of 1.5 mm for cost-effectiveness. Oblique incidence has been tested up to an angle of 60${}^{\circ }$.

A tunable FSS with circular loops connected by varactor diodes has been proposed in [15]. Depending on the bias of the diodes, the operational frequency varied from 0.54 to 2.50 GHz, with a stop-band width of 1.28 GHz, for a size of the square unit cell of 12 mm. The experimental board built on an FR4 substrate with a thickness of 1 mm has been tested up to an incidence angle of 60${}^{\circ }$, and showed a good polarization stability. In [16], an FSS with two distanced layers having different transmission zeros, with a pattern of cross dipoles with incurved arms has been considered. Miniaturization of the unit cell (7.7 × 7.7 mm${}^{2}$) allowed for good polarization stability and proper operation up to an angle of 60${}^{\circ }$ in the X-band, for a bandwidth of 3.59 GHz. Experiments have been carried over with a prototype comprising 21 × 14 elements and having Arlon DiClad 880 of 0.76 mm thickness for dielectric substrate.

A single-layer, single-sided solution having a very large –10 dB stop-band in the range 2.72–13 GHz, intended to be applied as a reflector to enhance the gain of UWB antennas has been reported in [17]. The pattern in the unit cell consisted of modified square rings. The experimental board has been fabricated on an FR4 substrate of 1.6 mm thickness and comprised 32 × 32 unit cell elements. A claimed 80${}^{\circ }$ angular stability has been reported. A single layer FSS with parallel convoluted elements patches printed on the opposite faces of a 1.6 mm thick FR4 substrate has been proposed in [18]. For dimensions of the unit cell of 10 × 10 mm${}^{2}$, a large –10 dB bandwidth in the range 3.1–13.3 GHz and an angle stability up to 45${}^{\circ }$ have been obtained. Experiments were performed with a prototype containing 30 × 30 unit cells.

In this paper, an FSS with a very large stop-band, designed to filter electromagnetic signals in the standardized UWB frequency range (3.1–10.6) GHz, is proposed. To ensure proper operation, the design provides a stop-band larger than the targeted one (a –10 dB stop-band is obtained between 1.75 and 15.44 GHz, spanning a 13.69 GHz wide bandwidth). The solution, which presents a stop-band much larger than the ones reported in the recently published papers, is implemented on a single-layered, double-face, cost-effective FR4 substrate, of 1.6 mm thickness. The unit cell extends over 15 × 15 mm${}^{2}$. The results of the numerical simulations have been efficiently verified by experiments, carried out on a board comprising 30 × 30 unit cells.

The rest of the paper is organized as follows. In the next section, the proposed solution is presented and the operation is assessed by electromagnetic simulation. Circuit models for the two faces of the unit cell and for the combination of the two are proposed and validated in Section 3. Experimental results are reported in Section 4 and conclusions are drawn in the last section.

## 2. Presentation of the Proposed Structure

The proposed periodic surface is built on a single-layer FR4 substrate ($\varepsilon_{r}$ = 4.3, tanδ = 0.025), of thickness $s_{t}$ = 1.6 mm, with metallization on both faces. The structure of the unit cell, of dimensions $d_{x}=d_{y}=15$ mm is reported in Figure 1.

The metal pattern on the upper face of the unit cell, referred to as “Face 1” (Figure 1a), consists of nine squares, having an edge $L_{2}$ = 2.2 mm and lines of width of $w_{1}$ = 1 mm and length $L_{1}$ = $d_{x}-d$ = 14.9 mm, where *d* = 0.1 mm is the distance between parallel square rings within adjacent unit cells. The distance between the centers of any two consecutive squares is $L_{1}$/3 in both *x* and *y* directions.

Face 2 (Figure 1b) contains four metal squares of edge $L_{sq}$ = 5 mm, with centers displaced by *T* = $d_{x}/2-L_{sq}$/2 − *d*/2 with respect to the center of the unit cell in both *x* and *y* directions and one hollow square, situated at the center of the unit cell, having an edge $L_{3}$ = 4.5 mm drawn with a line width $w_{2}$ = 0.1 mm.

The CAD model of the unit cell is reported in Figure 1c, with substrate removed to ensure visibility of the whole metallization. The conception of the structure relied on the interplay between the rather large dimension of the unit cell, of 15 mm, introduced to ensure an appropriate response to incident electromagnetic waves at low frequencies, and the presence of a certain number of small resonators in each unit cell, for providing a good response at higher frequencies. The resonators introduce several resonances that ensure the occurrence of a large stopband, as presented and explained next.

The structure has been firstly tested by simulation, in normal incidence of plane waves, with the E field parallel to the $d_{y}$ edge of the unit cell of Figure 1a, a situation called TE or *S* polarization [19]. The frequency domain solver, with periodic boundary conditions in the *x* and *y* directions has been used, and Floquet ports in the *z* direction have been defined. In the simulations, the substrate material has been defined as “FR4 (lossy)”. The TM (*P*) polarization yields the same results in normal incidence, due to the symmetry of the structure of the unit cell. The simulated transmittance (magnitude of $S_{21}$ in (dB)) is reported in Figure 2 (the blue, solid line curve). The measured transmittance is also represented in the same figure, with a red, dotted line for reference. The experimental setup and measuring procedure will be addressed in Section 4.

Figure 2 reveals the presence of a very large –10 dB stop-band, between 1.59 and 15.76 GHz, spanning a 14.17 GHz wide bandwidth (163.3% with respect to the central frequency). The simulated values for the stop-band limits have been confirmed by the measured ones, of 1.75 and 15.44 GHz, respectively. Therefore, the periodic surface acts like a band-stop filter with an ultra-wide bandwidth.

The proposed filter has been devised by first coupling two resonators with complementary frequency responses, as shown in Figure 3. The transmittance of a periodic surface having the metal pattern on one side only, namely Face 1 in Figure 1a, is displayed with a red, solid line in Figure 3. The transmittance presents a large stop-band around a resonant frequency of 6.87 GHz. This resonance is introduced by the dipoles of length $L_{1}$, as the field image of the surface current density on Face 1, in TE normal incidence, represented in Figure 4a shows (metallization on Face 2 was absent when the field image in Figure 4a was obtained by simulation).

Reactive loading of three of the five dipoles present on the unit cell provides the possibility to obtain a larger bandwidth. This has been used to optimize the dimensions of the dipoles and squares to obtain a large bandwidth. Note that the current flowing on the orthogonal direction (*x* in this case) is very small, so that the cross-pol coupling is also small.

As it can be seen in Figure 3, the periodic surface with metallization on Face 2 has only two resonance frequencies at 11.58 and 15.06 GHz. The corresponding field images are displayed in Figure 4b,c, respectively. Note that the resonance at 11.58 GHz is mainly determined by the hollow rectangles, while the one at 15.06 GHz is mainly determined by the capacitive coupling between the metal squares. When metallizations are present on both sides of the periodic structure, Figure 1c, the resulting periodic surface has resonant frequencies at 6.74 and 11.59 GHz, which are very close to the resonant frequencies of the structures based on Face 1 or Face 2 only. However, the resonance around 15 GHz is no more present. The field image in Figure 4d confirms that the operation of the filter is mainly determined by the pattern of Face 1 at 6.74 GHz.

A similar image (not reported) demonstrates a similar operation of the FSS at the other resonance frequency. Therefore, by coupling two band-stop filters, one with a large bandwidth at low frequencies and another one with a two-band structure at higher frequencies, a band-stop filter with an ultra-wide band has been obtained. Optimization was possible by considering the impact of various geometrical parameters on the frequency response.

The transmission and reflection coefficients in co- and cross-polarization are represented in Figure 5. The linearly polarized wave is incident from the side of Face 1. The same figure displays the phase of the reflection coefficient in co-polarization. The almost linear variation of the phase of the reflection coefficient with frequency is similar to the result reported in [13]. To gain a deeper insight into the operation of the proposed FSS, a circuit model is considered in the next section.

## 3. Circuit Model

Understanding the frequency response of an FSS is improved by circuit models, which account for resonances introduced by the structure that affect electromagnetic waves in normal incidence [20]. In this section, we present circuit models for the unit cells on the two faces of the proposed FSS individually, and for the unit cell of the FSS resulted by putting the two faces together.

The circuit model for Face 1 is presented in Figure 6a. The incident wave is modeled by a matched voltage generator connected at the input terminals, having an internal impedance equal to the wave impedance $Z_{0}$ of the free space. The FSS is represented by the combination of the shunt impedance *Z*, accounting for the metal pattern, and the transmission line of length $s_{t}$, accounting for the substrate. The characteristic impedance of the transmission line is denoted by $Z_{c}$, and the propagation constant is denoted by γ. Both quantities are determined by the material properties of the substrate (FR4). These parameters are calculated as follows:(1)$$\begin{array}{c} Z_{c}=\sqrt{\frac{\mu_{0}}{\varepsilon_{0}\varepsilon_{r}\left(1-ȷ\tan\delta \right)}} \end{array}$$
(2)$$\begin{array}{c} \gamma =ȷ\omega \sqrt{\mu_{0}\varepsilon_{0}\varepsilon_{r}\left(1-ȷ\tan\delta \right)} \end{array}$$
where $\mu_{0}$ represents the value of the absolute permeability in vacuum and $\varepsilon_{0}$ represents the value of the absolute permittivity in vacuum.

A matched load equal to the free space impedance is connected to the output terminals.

The $S_{21}$ parameter of the two-port in Figure 6a is given by:(3)$$S_{21}=\frac{2z_{p}}{1+z_{p}}\frac{1+\Gamma }{e^{\gamma d}+\Gamma e^{-\gamma d}}$$
where
(4)$$\begin{array}{c} \Gamma =\frac{Z_{0}-Z_{c}}{Z_{0}+Z_{c}} \end{array}$$
(5)$$Z_{0}=\sqrt{\frac{\mu_{0}}{\varepsilon_{0}}}$$
and
(6)$$z_{p}=\frac{zz_{in}}{z+z_{in}};\;$$
(7)$$z=\frac{Z}{Z_{0}};\;$$
(8)$$z_{in}=\frac{Z_{c}}{Z_{0}}\frac{Z_{0}+Z_{c}\tanh\left(\gamma d\right)}{Z_{c}+Z_{0}\tanh\left(\gamma d\right)}$$

The circuit model for Face 2 is obtained from the schematic in Figure 6a by feeding the two-port at the terminals from the right and terminating the left port by a matched load. However, Equations (Equation 3)–(Equation 8) for calculating the $S_{21}$ parameter still apply, due to reciprocity.

Both faces of the FSS introduce resonances, as reported in Figure 3. Since the resonances are widely separated in frequency, the shunt impedance *Z* in Figure 6a consists of several uncoupled resonant circuits, as shown in Figure 6b. The presence of the dissipating elements $R_{i}$ in Figure 6b is motivated by the losses introduced by the FR4 substrate in near field. The propagation of the waves in the substrate also introduces losses. However, propagation losses are negligibly small, as calculation of $S_{21}$ performed by alternatively considering or neglecting losses in the transmission line of very small length $s_{t}$ (the thickness of the substrate) in Figure 6a reveals.

To estimate the order of magnitude of the circuit elements, well-known approximation formulas can be considered as a starting point. The inductance *L* of a thin metal strip of length *s* and width *w* is given by
(9)$$L=\mu_{0}\frac{s}{2\pi }\ln\left(\frac{2s}{w}\right)$$

Then, the capacitance *C* of the resonant circuit can be calculated from the resonant frequency $f_{r}$ according to:(10)$$f_{r}=\frac{1}{2\pi \sqrt{LC}}$$

For example, the calculated inductance of a strip having *s* = 5 mm and *w* = 1 mm, belonging to the strip resonator in Figure 4a, is 2.28 nH. For *s* = 4.5 mm and *w* = 0.1 mm, corresponding to the edge of the hollow rectangle that bears a high current density in Figure 4b, the calculated inductance is 4.04 nH.

Equations (Equation 3)–(Equation 8) have been implemented in a Matlab${}^{\mathrm{TM}}$ script and the obtained transmittance has been compared with the one obtained by simulation. The values of the circuit elements have been found in a few iterations. The values obtained for the two values of the inductance mentioned above have been 0.62 and 3.29 nH, respectively. The resistances in the circuits have the role to ensure an appropriate Q value.

The circuit elements for the two faces are listed in Table 1. The second column contains the circuit elements and resonant frequencies associated with the unit cell of Face 1. The third column contains the same elements associated with Face 2. Three resonant frequencies, denoted $f_{ri}$, have been considered for Face 1 (*n* = 3 in Figure 6b) and four resonant frequencies for Face 2 (*n* = 4 in Figure 6b). Note that the frequency $f_{r4}$ lies outside the operating frequency range of the FSS; however, simulations have indicated its presence. This resonant frequency impacts the frequency response of the FSS in the operational frequency range.

The transmittance obtained with the circuit model for Face 1 is represented in Figure 7a, together with the transmittance obtained by simulation with CST for comparison. A similar representation for Face 2 is reported in Figure 7b. Both results indicate a good match.

When the two faces are put together to form the FSS, various couplings between constituting elements occur. These couplings are intricate and hard to identify individually, due to the large number of resonators that are present in the unit cell. The effect of the coupling can be referred to the input or the output of the transmission line that is present in the circuit model, or to both.

We have adopted the solution to refer the effect of coupling to the input, such that circuit elements and resonances are modified, and additional resonances might be present at the input (Face 1) due to the impact of the output (Face 2). This approach is motivated by the fact that the large resonance that occurs above 6 GHz when only Face 1 is present in the simulation (see Figure 3) also occurs when both faces are present, as revealed by the simulation and measurement results reported in Figure 2.

The selected circuit model has been therefore the one represented in Figure 6a, with the impedance *Z* having the structure in Figure 6b. In this case, *n* = 5 resonances had to be considered, such that five series RLC circuits connected in parallel entered the structure of the impedance *Z*.

Equations (Equation 1)–(Equation 8) have been used again to calculate the $S_{21}$ parameter of the circuit, and several iterations have been performed to calculate the values for the parameters, knowing the orders of magnitude from (Equation 9) and (Equation 10). The best fit has been obtained for the resonance frequencies and parameter values listed in the last column from Table 1. The transmittance obtained with the circuit model is represented in Figure 8. The transmittance obtained by simulation with CST is also reported in the same figure, for comparison. It can be seen that the transmittance calculated with the circuit model matches the one obtained by simulation in a reasonable way in the –10 dB stop-band, which is the frequency band of interest. Moreover, the band limits are correctly assessed by the circuit model.

## 4. Experimental Validation

The proposed periodic structure has been realized as a prototype of FR4 printed circuit board (PCB) comprising 30 unit cells in each of the two orthogonal directions for a total extension of 450 × 450 mm${}^{2}$. A photograph of the Face 1 of the prototype in presented in Figure 9a, whilst Face 2 is visible in Figure 9b. Measurements have been performed in an anechoic chamber, Figure 9c, by means of the same substitution method and equipment described in [21].

The transmittance of the prototype built on PCB has been measured in a full anechoic chamber, of dimensions 6 × 3 × 3 m${}^{3}$, having the metal walls, ceiling, and floor covered with absorbing pyramids working for frequencies higher than 1 GHz. The substitution method has been applied, by using a Keysight N5227A vector network analyzer (VNA) and two types of R&S HF906 double-ridged horn antennas. The distance between the two horn antennas was 2.8 m (the measured object being placed half-way and rotated around an axis passing through the middle of the structure).

First, the $S_{21}$ parameter of the system formed by the emitting and receiving horns, connected to the VNA has been measured in the case when the antennas have been separated by a tinfoil covered plywood having an empty window for inserting the sample, Figure 9c. Then, the measurements have been repeated with the sample inserted, Figure 9d. The transmittance of the sample in dB could be calculated by taking the difference of the results from the second and first measurements described above. A simple mechanism has been devised for allowing the rotation of the screen at prescribed angles in view of measuring the transmittance at different incidence angles. It consists of a fix pedestal and a rotating panel with the hole (mentioned above with reference to Figure 9c). The extension of the tinfoil covered plywood is the maximum available for the normal incidence, corresponding to an infinite ground plane. When rotated, some space between it and the tips of the cones appear, but the diffraction from the edges are cancelled by the calibration process (difference method described above). The accuracy is of 1${}^{\circ }$, but, during the measurements, a 5${}^{\circ }$ step sequence has been considered, since the rotation has been performed by hand.

The measured transmittance in normal incidence is represented in Figure 2 with a red, dotted curve, together with the simulated one, in order to reveal the good matching between the two. The simulated transmittance for TE incidence for colatitudes (angle θ in spherical coordinates) between 0 and 60${}^{\circ }$ and the measured ones are reported in Figure 10.

Measurements have been limited to an angle of incidence of 60${}^{\circ }$ as the effective aperture for the incident waves decreases by 50% with respect to normal incidence. The same curves but for TM incidence are represented in Figure 11. This set of results is sufficient for assessing the variability of the transmittance at oblique incidence, due to the symmetry of the metal pattern in the unit cell.

The reported data show a good agreement between simulation and measurement results. Some minor differences exist for small values of the transmittance, i.e., in the stop-band that are caused by variations in the geometry and irregularities in the dielectric of the PCB, by tolerances in the metallization and by higher order modes that are launched as surface waves, which can radiate when reaching the bounds of the PCB. The last phenomenon might be significant at some frequencies only, for large angles of incidence.

Possible leakage of energy due to propagative modes must also be evaluated. At an angle of incidence of 15${}^{\circ }$, modes TE(–1,0) and TM(–1,0) become propagative at a frequency of 15.88 GHz. However, the transmittance is below –25 dB for all considered frequencies. The same modes become propagative at 13.33 GHz for θ = 30${}^{\circ }$, at 11.71 GHz for θ = 45${}^{\circ }$, and at 10.72 GHz for θ = 60${}^{\circ }$. However, the maximum values of the transmittance are –35 dB in the first two cases, and –40 dB in the last one, making the leakage due to propagative modes other than TE(0,0) and TM(0,0) negligibly small.

To illustrate the impact of higher order modes, we present in Figure 12 the magnitudes of the $S_{21}$ parameters corresponding to the modes listed in Table 2, for TE(0,0) incidence (Figure 12a) and TM(0,0) incidence (Figure 12b), at an angle θ = 30${}^{\circ }$ and azimuth ϕ = 0. The Rayleigh frequencies are also listed for convenience in Table 2. The results have been obtained by simulation with [19].

The reported results indicate that, indeed, the only propagating modes in the frequency range of interest are TE(−1,0) and TM(−1,0) for TE and TM incidence, respectively, and the corresponding transmittances are below –10 dB. Furthermore, evanescent modes have negligibly small transmittances in the operating frequency band of the FSS.

The transmittance of the structure has been further measured in TE and TM incidence for angles of incidence between 0 and 60${}^{\circ }$ with a 5${}^{\circ }$ increment and the stop-band has been determined in each case, since it is the most important parameter of the proposed filtering surface. Results are reported in Figure 13, the stop-band for the TE case being denoted by [$f_{LTE}$;$f_{HTE}$] and for the TM case by [$f_{LTM}$; $f_{HTM}$].

Results in Figure 13 reveal that the stop-band is practically constant in TE incidence up to an angle of incidence of 60${}^{\circ }$ ($f_{LTE}$ varies from 1.75 GHz at 0${}^{\circ }$ to 1.66 GHz at 60${}^{\circ }$ and $f_{HTE}$ increases from 15.44 GHz at 0${}^{\circ }$ to values above 20 GHz between 45${}^{\circ }$ and 60${}^{\circ }$). In the TM incidence case, the stability of the stop-band can be considered reasonable up to an angle of incidence of 50${}^{\circ }$ ($f_{LTM}$ increases from 1.75 GHz at 0${}^{\circ }$ to 2.87 GHz at 50${}^{\circ }$, 3.34 GHz at 55${}^{\circ }$ and 4.00 GHz at 60${}^{\circ }$, and $f_{HTM}$ varies from 15.44 GHz at 0${}^{\circ }$ to 17.78 GHz at 45${}^{\circ }$ and 17.13 GHz at 60${}^{\circ }$). Nevertheless, the stop-band remains ultra-wide for all angles of incidence.

A response that is more dependent on the angle in TM incidence than in TE incidence has been noticed in other works too [15]. This can be explained by considering the interaction of the electric and magnetic field intensities of the incident wave with metal dipoles and loops placed on the two sides of the PCB. In TE incidence, as the incidence angle θ increases, the length of the projection of the E vector on the surface remains constant and the projection of the H vector on the *z*-axis, which is perpendicular to the surface of the loops increases, and so does the interaction between the two.

On the other hand, in TM incidence, as the incidence angle θ increases, the projection of the E vector on the surface of the FSS decreases, making the interaction with the dipoles smaller, while the H vector remains parallel to the surface of the loops. The interaction of the H field with the FSS is achieved in this case through the loops that are perpendicular to the structure (parallel to the *z*-axis) that are closed by capacitive effect between the metal patches on the two faces of the FSS.

## 5. Conclusions

In this paper, an FSS operating as a band-stop spatial filter with a very large stopband, intended for filtering in the UWB frequency range, with good polarization insensitivity and angle stability has been proposed. The structure has been implemented on a cost-effective FR4 substrate. A very large stopband has been obtained, in the range of 1.75–15.44 GHz. This compares favorably with other works reported in the literature having similar targets. Table 3 contains a comparison with similar works ($\lambda_{c}$ is the free-space wavelength corresponding to the mid-band frequency, and $\lambda_{0}$ corresponds to the lower frequency of the stopband).

The first line summarizes an older solution, which raised the interest in wideband FSSs. The next two lines are from works dedicated to filtering in the UWB frequency range, similarly to the proposed solution. The last line reports parameters of the solution proposed in this paper. The largest bandwidth has been obtained in our case. However, the necessity to decrease the lower bound of the stopband imposed a larger dimension for the unit cell, which somehow impacted the angle stability, which is lower than the one reported in [17]. As explained in Section 2, for the correct operation at higher frequencies, smaller resonators have been inserted in the unit cell.

The proposed structure has been assessed by simulation, circuit model and measurement in an anechoic chamber. The obtained results demonstrated a good agreement between theory and experiments.

## Figures and Tables

**Figure 1 sensors-22-01896-f001:**
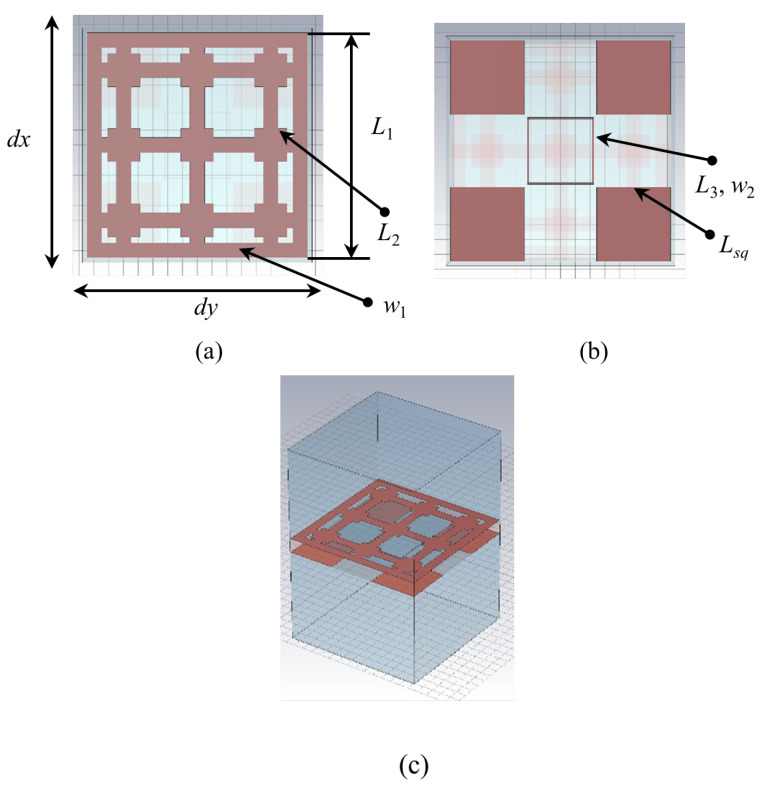
Unit cell geometry—CAD model: (**a**) Face 1; (**b**) Face 2; (**c**) rendering with removed substrate for better visibility.

**Figure 2 sensors-22-01896-f002:**
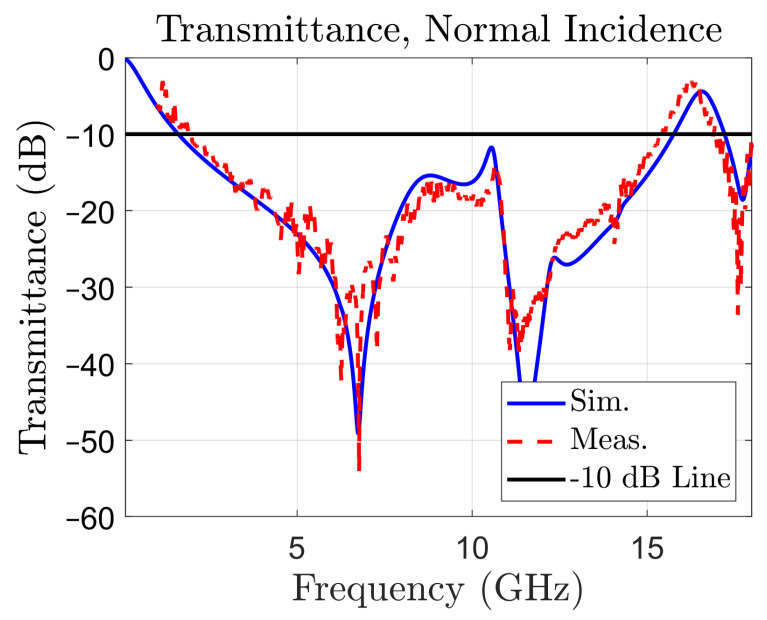
Transmittance of the periodic surface in normal incidence: simulated (blue, solid line), and measured (red, dotted line). The –10 dB line is used to determine bandwidth.

**Figure 3 sensors-22-01896-f003:**
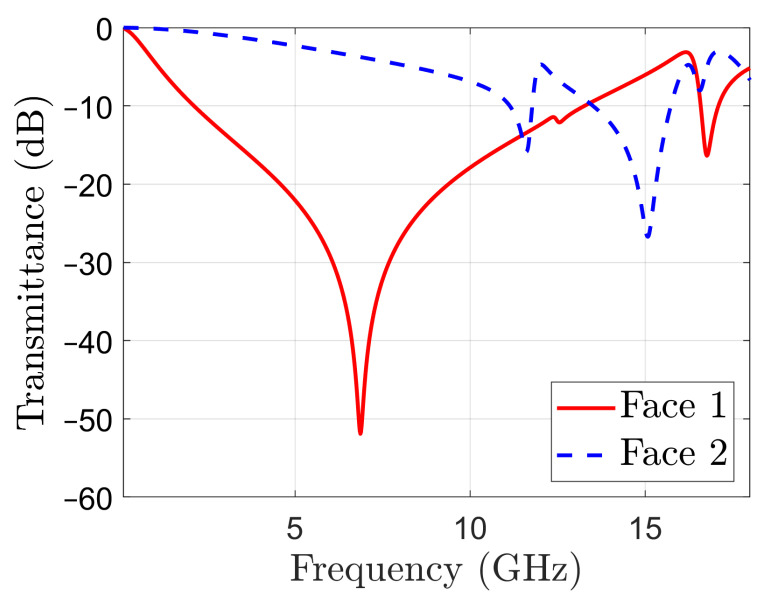
Transmittance in normal incidence of periodic surfaces with metal patterns on one face only, as in Figure 1: Face 1 solid line; Face 2 dotted line.

**Figure 4 sensors-22-01896-f004:**
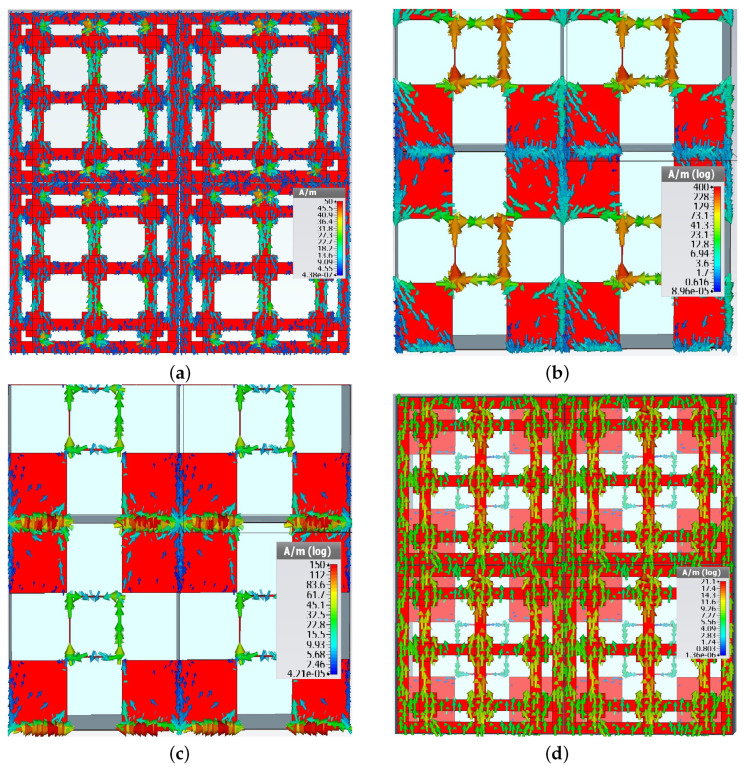
Field images of surface current density for the periodic surface with: (**a**) Face 1 only present, at 6.87 GHz; (**b**) Face 2 only present, at 11.58 GHz; (**c**) Face 2 only present, at 15.06 GHz. (**d**) View of Face 1, when both metallizations are present, at 6.74 GHz. The use of 2 × 2 configuration is preferred since it allows a better appreciation of the field distribution between adjacent unit cells.

**Figure 5 sensors-22-01896-f005:**
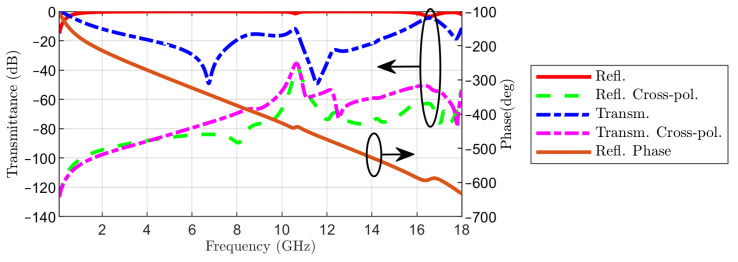
Reflection and transmission coefficients in co- and cross-polarization and phase of reflection coefficient in co-polarization, when the linearly polarized wave is incident from Face 1 of the FSS.

**Figure 6 sensors-22-01896-f006:**
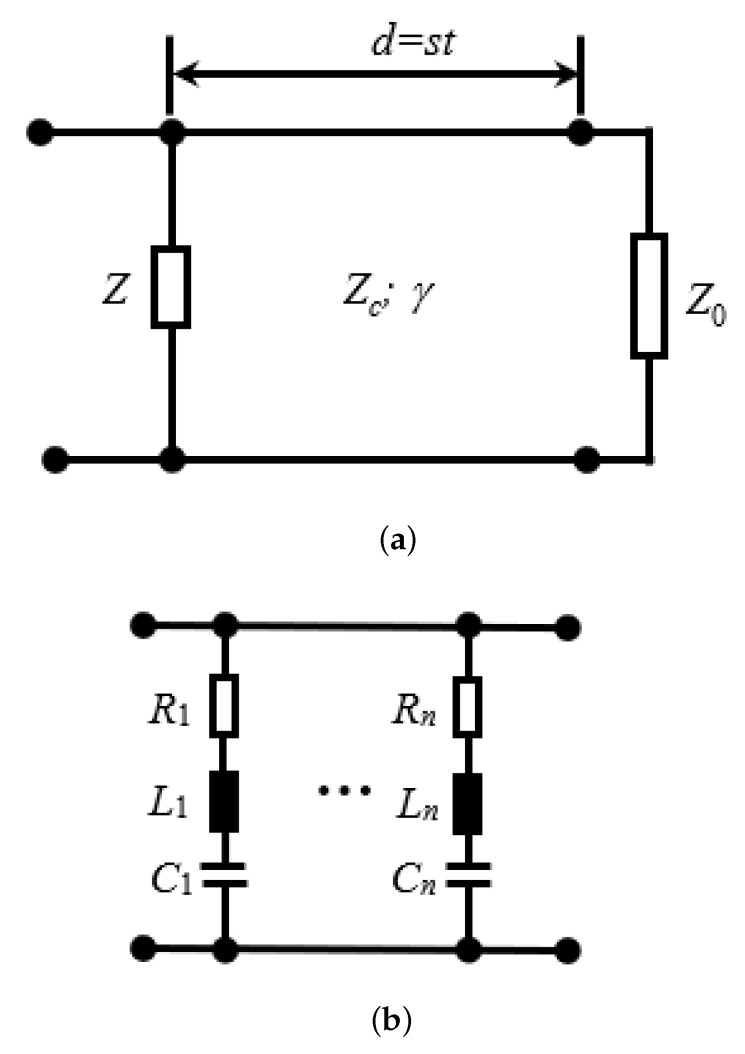
Equivalent circuits: (**a**) unit cell; (**b**) impedance *Z*.

**Figure 7 sensors-22-01896-f007:**
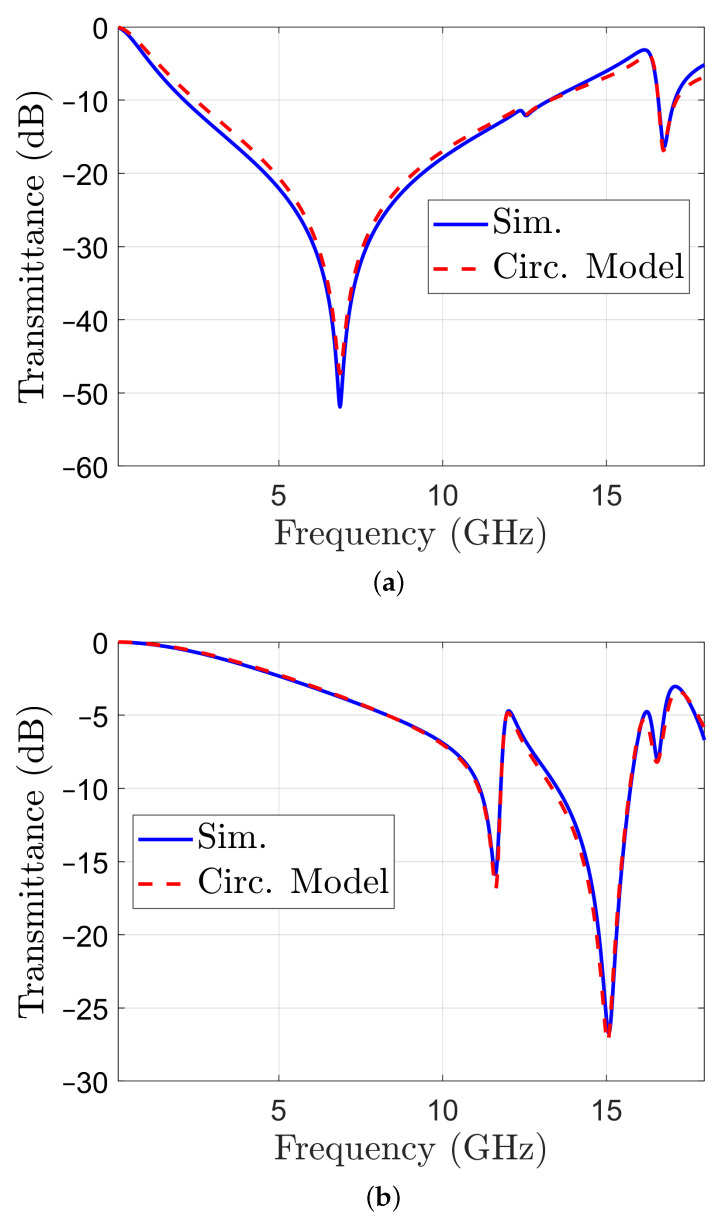
Transmittance of structure with metal pattern on one face only: calculated with the circuit model of the unit cell (dashed line) and obtained by simulation (solid line): (**a**) Face 1; (**b**) Face 2.

**Figure 8 sensors-22-01896-f008:**
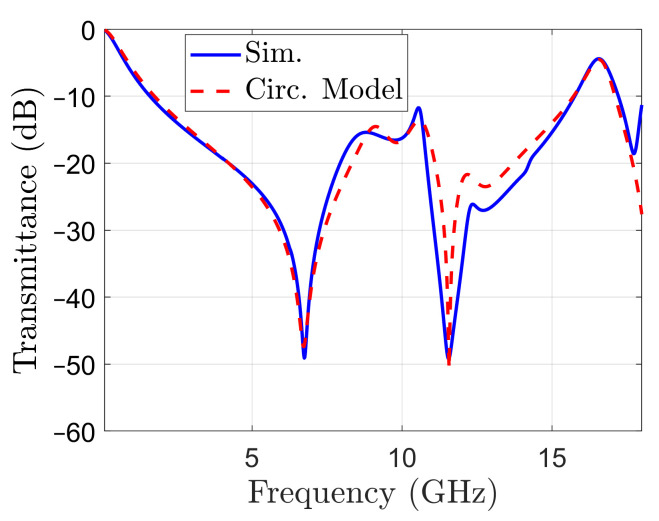
Transmittance of the full structure: calculated with the circuit model of the unit cell (dashed line) and obtained by simulation (solid line).

**Figure 9 sensors-22-01896-f009:**
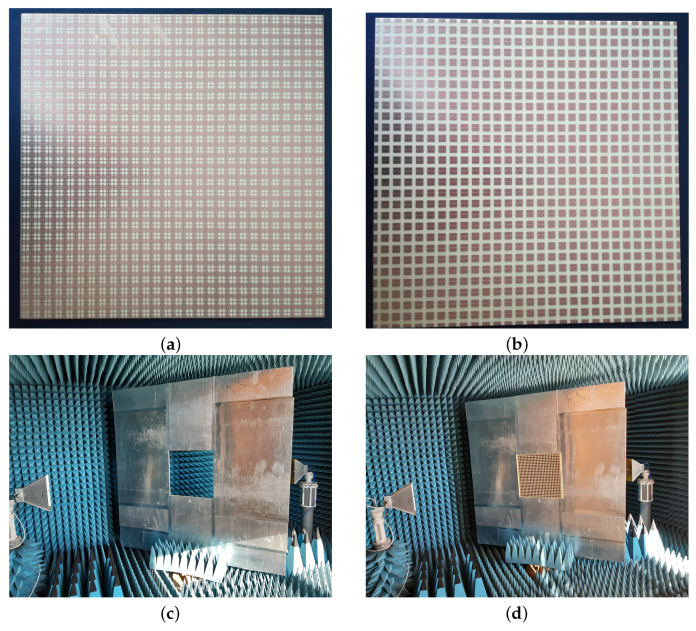
Experimental validation: PCB prototype Face 1 (**a**), and Face 2 (**b**); measurement setup without (**c**), and with (**d**) prototype inserted.

**Figure 10 sensors-22-01896-f010:**
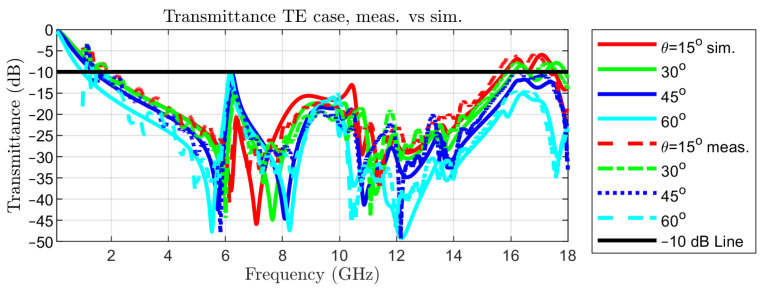
Transmittance in oblique incidence of TE waves: simulated vs. measured.

**Figure 11 sensors-22-01896-f011:**
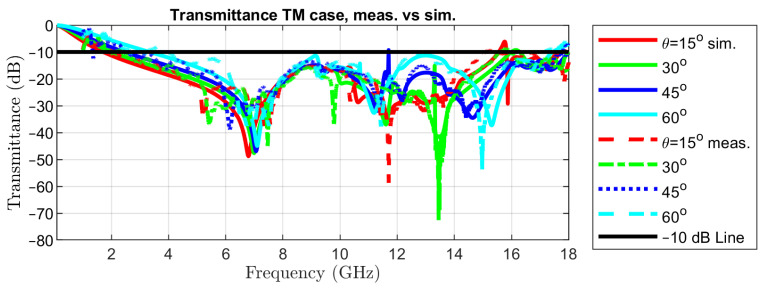
Transmittance in oblique incidence of TM waves: simulated vs. measured.

**Figure 12 sensors-22-01896-f012:**
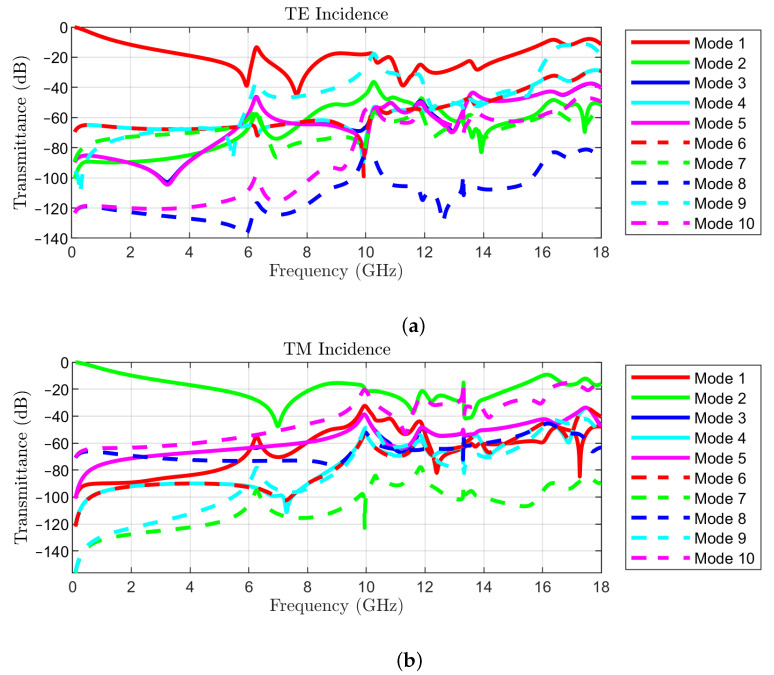
Transmittances of 10 modes for θ = 30${}^{\circ }$: (**a**) TE incidence of mode 1; (**b**) TM incidence of mode 2.

**Figure 13 sensors-22-01896-f013:**
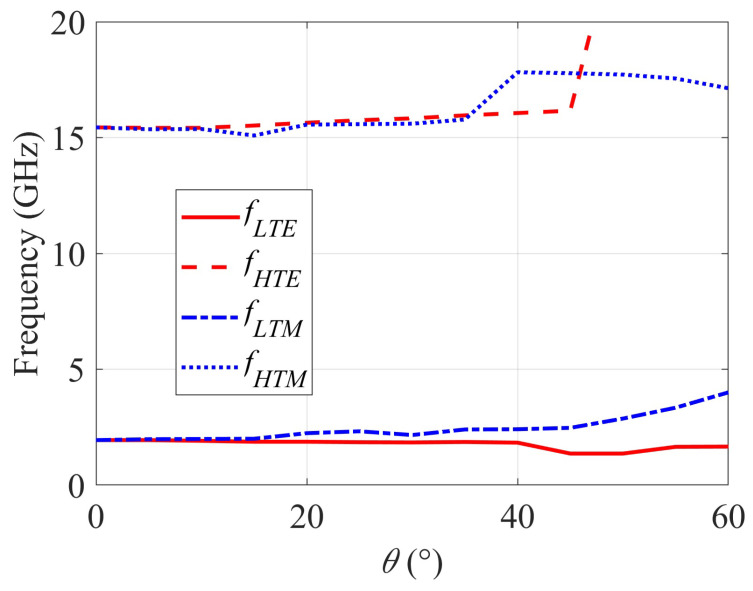
Stop-band limits versus incidence angle.

**Table 1 sensors-22-01896-t001:** Elements of the circuit models for the unit cells.

	Face 1	Face 2	FSS
$f_{r1}$ (GHz)	6.87	11.65	6.72
$f_{r2}$ (GHz)	16.73	15.05	9.80
$f_{r3}$ (GHz)	12.48	16.55	11.57
$f_{r4}$ (GHz)	-	20.34	12.63
$f_{r5}$ (GHz)	-	-	18.3
$R_{1}\left(\Omega \right)$	0.74	27.52	0.74
$R_{2}\left(\Omega \right)$	20.73	5.65	32.04
$R_{3}\left(\Omega \right)$	263.89	94.25	0.42
$R_{4}\left(\Omega \right)$	-	0.42	13.31
$R_{5}\left(\Omega \right)$	-	-	0.4
$C_{1}$ (fF)	860	8.80	980
$C_{2}$ (fF)	5.50	34.0	75.5
$C_{3}$ (fF)	1.00	2.80	85.0
$C_{4}$ (fF)	-	41.0	105.0
$C_{5}$ (fF)	-	-	75.0
$L_{1}$ (nH)	0.62	21.21	0.57
$L_{2}$ (nH)	16.45	3.29	3.49
$L_{3}$ (nH)	162.63	33.03	2.23
$L_{4}$ (nH)	-	2.26	1.51
$L_{5}$ (nH)	-	-	1.01

**Table 2 sensors-22-01896-t002:** Mode numbering for incidence at θ = 30${}^{\circ }$ (ϕ = 0) and corresponding Rayleigh frequencies.

Mode nr.	1	2	3	4	5	6	7	8	9	10
Mode	TE	TM	TE	TM	TE	TM	TE	TM	TE	TM
	(0,0)	(0,0)	(0,1)	(0,1)	(0,−1)	(0,−1)	(1,0)	(1,0)	(−1,0)	(−1,0)
Rayleigh frequency (GHz)	0	0	23.08	23.08	23.08	23.08	39.97	39.97	13.32	13.32

**Table 3 sensors-22-01896-t003:** Comparison with other works.

	Stop-Band (GHz)	*d*/$\lambda_{c}$	*d*/$\lambda_{0}$	Angle Sensitivity	Substrate Type	Substrate Thickness (mm)
[13]	6.5–14	0.41	0.26	45${}^{\circ }$	FR4	3.2
[17]	2.72–13.23	0.21	0.07	80${}^{\circ }$	FR4	1.6
[18]	3.1–13.3	0.27	0.1	45${}^{\circ }$	FR4	1.6
Present work	1.75–15.44	0.43	0.09	60${}^{\circ }$ TE, >50${}^{\circ }$ TM	FR4	1.6

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
