# Peer review of "Frequency Selective Surface for Ultra-Wide Band Filtering and Shielding"

_sensors, 2022, doi:10.3390/s22051896_

Round 1

Reviewer 1 Report

The authors presented a frequency selective surface for spatial filtering in the UWB frequency range. A wide stop-band ranging from 1.75 to 15.44 GHz has been obtained with good polarization insensitivity and an angular stability of more than 50-degrees.The work has enough contribution, however it needs some improvements.

  1. The presentation of the data should be improved, especially Figure 11. Plot in the same style as other graphs.
  2. Latest papers should be added in references to locate this work in state of the art work. Consider to the following papers published in MDPI journals: https://doi.org/10.3390/app8091689, https://doi.org/10.3390/app10249125, https://doi.org/10.3390/mi12091027
  3. Check equations, their numbering has been misplaced. Formate manuscript carefully in the revised manuscript.
  4. If possible, add reflection phase characteristics of the FSS as shown in [11].
  5. Comment on the usage of FR-4 substrate at high frequencies, as the FSS works upto 15 GHz. The dielectric properties of the FR-4 changes significantly at higher frequencies.

Reviewer 2 Report

The authors present a sound FSS design. It would be very convenient, with respect to Figures 9 and 10, that experimental results are shown superimposed to simulated results, to actually check the level of agreement (i.e., at each incident angle). 

With regard to the impact of higher-order modes, in view of Table 2, where it can be seen how there are propagating higher-modes in the considered frequency band (i.e. < 18 GHz), I am wondering if in each and every simulation the authors have ensured that no propagation of higher-order modes appear. The claim that "some minor differences...are caused... and by higher order modes that are launched as surface waves" should also be better explained, as I think CST Studio should be able to predict the behaviour due to this phenomenon in its full-wave simulation (not the radiation at the FSS boundary, but the appearance of surface waves).
